# Glucocorticoid Nanoparticles Show Full Therapeutic Efficacy in a Mouse Model of Acute Lung Injury and Concomitantly Reduce Adverse Effects

**DOI:** 10.3390/ijms242316843

**Published:** 2023-11-28

**Authors:** Gesa J. Albers, Agathe Amouret, Katrin Ciupka, Elena Montes-Cobos, Claus Feldmann, Holger M. Reichardt

**Affiliations:** 1Institute for Cellular and Molecular Immunology, University Medical Center Göttingen, 37073 Göttingen, Germany; 2Institute of Inorganic Chemistry, Karlsruhe Institute of Technology, 76131 Karlsruhe, Germany; claus.feldmann@kit.edu

**Keywords:** acute lung injury, glucocorticoids, nanoparticles, myeloid cells, alveolar type II cells, muscle atrophy

## Abstract

Glucocorticoids (GCs) are widely used to treat inflammatory disorders such as acute lung injury (ALI). Here, we explored inorganic–organic hybrid nanoparticles (IOH-NPs) as a new delivery vehicle for GCs in a mouse model of ALI. Betamethasone (BMZ) encapsulated into IOH-NPs (BNPs) ameliorated the massive infiltration of neutrophils into the airways with a similar efficacy as the free drug. This was accompanied by a potent inhibition of pulmonary gene expression and secretion of pro-inflammatory mediators, whereas the alveolar–capillary barrier integrity was only restored by BMZ in its traditional form. Experiments with genetically engineered mice identified myeloid cells and alveolar type II (AT II) cells as essential targets of BNPs in ALI therapy, confirming their high cell-type specificity. Consequently, adverse effects were reduced when using IOH-NPs for GC delivery. BNPs did not alter T and B cell numbers in the blood and also prevented the induction of muscle atrophy after three days of treatment. Collectively, our data suggest that IOH-NPs target GCs to myeloid and AT II cells, resulting in full therapeutic efficacy in the treatment of ALI while being associated with reduced adverse effects.

## 1. Introduction

Acute lung injury (ALI) is caused by disruption of the alveolar–capillary membrane integrity followed by excessive neutrophil migration into the alveoli [1]. Increased endothelial permeability permits the efflux of protein-rich fluid into the interstitium and its translocation into the alveolar space. This process is facilitated by additional injury to the epithelium, which is composed of flat alveolar type I (AT I) cells and cuboidal AT II cells [2,3]. Damage of the alveolar–capillary barrier then promotes the infiltration of neutrophils, which release detrimental pro-inflammatory, pro-apoptotic and cytotoxic mediators, and also trigger deployment of neutrophilic extracellular traps (NET) [4,5]. Inflammation is potentiated by resident pulmonary macrophages that release various chemokines such as MCP, thereby promoting additional recruitment of neutrophils and monocytes from the circulation into the damaged lung.

There is an urgent need for new therapeutic strategies to treat ALI. Clinical trials involving nitric oxide, surfactant replacement and β2-adrenergic agonists failed to reduce mortality, and studies assessing the therapeutic potential of anti-inflammatory drugs such as prostaglandin E1, lisofylline and GM-CSF were also disappointing [6,7]. Glucocorticoid (GC) therapy is controversial too, albeit about 20% of all patients worldwide are treated with it [6,8]. As a matter of fact, some clinical trials unveiled improved survival after GC administration, whereas others failed to demonstrate a better outcome [9,10]. Recently, evidence for a beneficial effect of GCs in ALI has also been obtained in COVID-19 patients [11]. Results from the RECOVERY study, which enclosed more than 10,000 patients, indicated that dexamethason (Dex) application diminished the 28-day mortality rate in patients receiving invasive mechanical ventilation [12]. This finding was later confirmed by a meta-analysis, albeit timing and dosing of GCs remain a matter of debate [13].

While GCs are naturally produced in the adrenal gland in response to stressful events, a plethora of synthetic derivatives with improved pharmacological properties have been developed for the treatment of inflammatory diseases [14]. GCs bind to their receptor (GR) located in the cytosol, which then induces or represses gene expression following nuclear translocation [15]. Consequently, the production of cytokines and inflammatory mediators is inhibited, T cells undergo apoptosis, and leukocyte migration is altered [16,17,18,19,20]. However, not only immune cells, but also other cell types involved in inflammatory lung diseases such as AT II and endothelial cells are targets of GCs [21]. Moreover, GC activities outside of the immune system can cause adverse effects collectively known as the iotrogenic Cushing’s syndrome [22]. In particular, the protracted application of high doses of GCs can induce osteoporosis, muscle atrophy and hypertension [23,24]. As a matter of fact, the most serious complication of GC application in ALI patients is myopathy, which severely compromises the overall benefit of this therapy and may also explain the ambiguous results on mortality obtained in clinical trials [10].

Mouse models have considerably contributed to elucidate the mechanisms of GCs in ALI therapy [25]. In one study, administration of methylprednisolone (MP) was shown to mitigate the disease by inducing the M2 polarization of macrophages [26], whereas another study unraveled reduced pulmonary inflammation and alveolar–capillary barrier permeability following Dex treatment. Using genetically engineered mice, the crucial role of SphK1 in macrophages was identified for the therapeutic activity of GCs, contributing to the integrity of the alveolar–capillary barrier [27]. Moreover, an important function of the dimeric GR in ALI mice was found under intensive care conditions [28]. Overall, these results underscore the great potential of GCs in ALI therapy.

Considerable efforts are being made to reduce the side-effects of GC therapy, which can be achieved with nanostructured carrier systems [29,30,31,32]. The unfavorable activities of traditional GCs are often related to their uniform tissue distribution and short half-life in the blood. Hence, new delivery vehicles have been developed including liposomes, polymer–drug conjugates, and inorganic scaffolds to overcome these hurdles. Following the same strategy, we have developed inorganic–organic hybrid nanoparticles (IOH-NPs) encapsulating GCs [33]. Precipitation upon the combination of zirconyl ([ZrO]^2+^) as an inorganic cation with a mixture of betamethasone phosphate ([BMP]^2−^, 90 mol-%) as the drug anion and flavinmononucleotide ([FMN]^2−^, 10 mol-%) as the fluorecent anion results in the stable formation of [ZrO]^2+^[(BMP)_0.9_(FMN)_0.1_]^2−^ IOH-NPs (designated BNPs in the following), which display a hydrodynamic diameter of 30–40 nm and excellent colloidal stability. Further analyses revealed that BNPs are characterized by a high cell type specificity and exert potent immunomodulatory activity both in vitro and in vivo [33,34,35,36].

In the current work, we compared BNPs to free betamethasone (BMZ) in their ability to suppress clinical symptoms and molecular hallmarks in a mouse model of ALI. Genetically engineered mice served to explore the cellular mechanism of BNPs. In addition, an induction of adverse effects was compared between BMZ and BNPs. Our findings unveil that BNPs mitigate ALI with a similar efficacy to BMZ, despite selectively targeting myeloid and AT II cells. As a consequence, application of BNPs does not affect T and B cell numbers in the blood, prevents the induction of muscle atrophy and thereby increases tolerability.

## 2. Results

### 2.1. Synthesis, Characterization and Tissue Distribution of BNPs

BNPs were developed as part of a platform concept of novel nanomaterials for drug delivery, and combine a simple synthesis in water, an extraordinary high drug load (>60% of total nanoparticle mass), an uncomplex composition and structure of the nanocarriers, and high adaptability of the IOH-NP concept to use various drugs. The synthesis of BNPs and their characterization have been published previously [33]. A brief summary of selected analyses of particle size, size distribution, zeta-potential, chemical composition and spectral features is provided in Appendix A. To confirm that BNPs reach the lung following in vivo administration, we determined the concentration of Zr, the inorganic component of BNPs, in several organs as a surrogate of IOH-NP distribution. Mice received BNPs via an i.p. injection, and 5 h later, they were sacrificed (Appendix A). The lung, small intestine and liver were dissected and analyzed for their Zr content using inductive coupled plasma mass spectrometry (ICP-MS) [37]. Hereby, the presence of BNPs could be demonstrated in all three organs, with a slightly higher concentration in liver (Appendix A). We conclude that BNPs reach the lung and thus represent a suitable GC nanoformulation to treat ALI.

### 2.2. GC Treatment Prevents Hypothermia in a Mouse Model of Sepsis

Bacterial sepsis is a frequent source of ALI [38]. Hence, we initially employed a model of endotoxin-induced sepsis in C57BL/6 mice and analyzed the impact of GC therapy on physiological hallmarks (Figure 1A). Within 15 h after LPS injection, the body weight of the mice was reduced, accompanied by a drop in body temperature and blood glucose levels (Figure 1B–D). GCs had no effect on body weight while both BMZ and BNPs administered at a dose of 10 mg/kg prevented the reduction in body temperature (Figure 1B,C). In addition, there was a trend that GCs attenuated hypoglycemia as well, albeit not reaching statistical significance (Figure 1D). Of note, there were no differences in therapeutic efficacy observed between BMZ and BNPs (Figure 1B–D).

### 2.3. Airway Inflammation in a Mouse Model of ALI Is Repressed by GCs

In the next step, we induced ALI in C57BL/6 wildtype mice via an i.p. injection of LPS followed by an i.v. injection of oleic acid (OA). Some mice were additionally treated with 10 mg/kg BMZ or BNPs to interrogate the impact of GCs and their formulation on ALI (Figure 2A). Occasionally, signs of breathing difficulty were already observed after 12 h, for which reason the health status was monitored throughout the last 3 h (Figure 2A). The experiment was terminated after 15 h when clinical symptoms of ALI were fully developed, and the BALF and lung, as well as the serum were collected (Figure 2A). Cells in the BALF were counted and stained with monoclonal antibodies recognizing surface molecules selectively expressed by macrophages (MΦ), monocytes (MO) and neutrophils (Appendix A). Cell numbers in the BALF of mice having received LPS and OA were significantly higher than those in control mice treated with PBS only (Figure 2B). BMZ and BNPs both diminished the cell number, but without any difference between both GC formulations. Flow cytometric analysis revealed strongly increased neutrophil numbers in the BALF of ALI mice, while the number of MΦ and MO was only moderately enhanced (Figure 2C,D). BMZ and BNPs suppressed airway neutrophilia with similar efficacy, but did not affect the abundance of MΦ and MO in the airways.

To confirm our flow cytometric results, we performed histological analysis via H&E staining of lung sections. Large bronchi and spacious alveoli separated by thin septae can be seen in the control mice (Figure 3). In contrast, alveoli in the mice treated with LPS and OA were densely filled with infiltrating cells, and the bronchi were constricted and displayed a thickened epithelial lining. These histological hallmarks of ALI were largely resolved in mice treated with BMZ or BNPs, indicating that GCs suppress the disease regardless of their formulation (Figure 3 and Appendix A).

Eventually, we investigated the gene expression of inflammatory mediators in the lung (Figure 4). After flushing the airways with PBS to remove infiltrating leukocytes, the mRNA levels of *Il6*, *Il1b*, *Tnf* and *Ccl2* were analyzed via RT-qPCR. The expression of all four genes was strongly elevated after induction of ALI, which was partially prevented by GC administration. There was a trend that BNPs were somewhat more efficient than BMZ, but without reaching significance (Figure 4).

### 2.4. GCs Improve Systemic Inflammation in a Mouse Model of ALI

ALI is not only characterized by local inflammation in the airways, but also involves a systemic cytokine storm. To address this issue, we analyzed the serum samples from mice subjected to ALI induction and GC treatment. Secretion of IL-6, IL-1β, TNFα and CCL2 was strongly elevated in mice with experimentally induced ALI, which was repressed by BMZ and BNPs with similar efficacy (Figure 5A). Analysis of the NO_2_^−^ concentration in the serum, which mirrors systemic NO production by myeloid cells, revealed an increase in ALI mice and a significant repression by BMZ, as well as BNPs (Figure 5B).

### 2.5. Differential Impact of GCs on the Integrity of the Alveolar–Capillary Barrier

The permeability of the alveolar–capillary barrier increases during ALI, thus promoting the leakage of liquid, proteins and inflammatory cells into the lung. This feature can be evaluated through an i.v. injection of Evans Blue (EB) dye and subsequently quantifying its amount having diffused into the lung tissue. The concentration of EB was increased in the lung of ALI mice, as expected, which was prevented by BMZ administration (Figure 5C). This finding confirms that GCs promote the maintenance of alveolar–capillary barrier integrity. Importantly, this was not the case for nanoformulated GCs, since BNP application had no effect on this feature.

### 2.6. Identification of BNP Target Cells in ALI Therapy

Free GCs reach all cell types with similar efficacy. In contrast, the effects of drugs encapsulated in IOH-NPs are highly specific because they primarily target cells with endocytic activity, among other macrophages and AT II cells [34]. Further inspired by previous studies revealing important roles of myeloid and AT II cells in the GC treatment of inflammatory lung diseases, we here tackled their involvement in BNP therapy of ALI [21,27,39]. To this end, two genetically engineered mouse strains were employed: GR^lysM^ mice lacking GR expression in MΦ, MO and neutrophils; and GR^spc^ mice lacking GR only in AT II cells.

BNPs did not prevent airway inflammation in GR^lysM^ mice, and neutrophil numbers in the BALF, as well as IL-6 serum levels remained unaltered after treatment of ALI mice (Figure 6A). Importantly, BNPs also failed to suppress leukocyte infiltration into the airways of GR^spc^ mice (Figure 6B). The total number of BALF cells and the number of neutrophils were similar regardless of GC therapy, indicating that the repression of local inflammation by BNPs requires GC activity in AT II cells too. In contrast, BNPs were still able to repress the secretion of IL-6 in GR^spc^ mice (Figure 6B), which confirms that the control of systemic inflammation by BNPs does not depend on GR expression in AT II cells.

### 2.7. Cell-Type-Specific Effects of BNPs on Blood Leukocytes

Previous analyses revealed that individual types of immune cell engulfed IOH-NPs with different efficacies in vitro [34]. To address this feature in vivo, mice were injected with LPS, treated with BMZ or BNPs, and 15 h later, blood samples were analyzed via flow cytometry. GC administration significantly increased the percentage of neutrophils, while the percentage of monocytes was diminished (Figure 7A). No difference in this respect was observed between the GC formulations. In contrast, the percentages of T and B cells in the blood were exclusively reduced by BMZ, but not BNPs (Figure 7A). These data confirm that BNPs selectively act via myeloid cells, while BMZ targets all leukocyte subsets similarly.

### 2.8. Impact of the GC Nanoformulation on the Induction of Muscle Atrophy

The treatment of ALI with GCs is considerably complicated by adverse effects, especially induction of myopathy. To evaluate whether BMZ and BNPs differ in this respect, we employed a mouse model of muscle atrophy. C57BL/6 mice received daily injections of BMZ, BNPs or PBS as the control, and on day 4, the tibialis anterior and gastrocnemius muscles were dissected. Administration of free GCs significantly reduced their weight, by approximately 20%, whereas muscle atrophy was prevented by the encapsulation of GCs into IOH-NPs (Figure 7B).

## 3. Discussion

ALI and its most severe form, acute respiratory distress syndrome (ARDS), are diseases of enormous societal impact [6] and diagnosed in about 10% of patients admitted to intensive care units worldwide. It is characterized by an extremely high mortality rate of around 40%, and even patients having survived the disease often require prolonged medical follow-up care [40]. Although some advance in the management of ALI has been made, especially by improving ventilation strategies, there are almost no pharmaceutical options available. Multiple drugs, including anti-inflammatory compounds, have been evaluated in clinical trials, but none of them are promising. GC administration is another option, although its efficacy is debatable [6,8]. The benefit of GCs in preventing ALI was evaluated, among others, in a clinical trial in which inhaled budesonide was combined with formoterol, leading to a halted progression to ARDS and improved oxygenation [41]. More frequently, however, GCs are used to treat already-established ALI. It is estimated that around 20% of all patients worldwide receive systemic GC administration to dampen the inflammatory response and restore alveolar–capillary barrier integrity. Nonetheless, evidence for therapeutic efficacy is somewhat ambiguous. One study which investigated the MP treatment of ALI failed to demonstrate any surviving advantage for patients having received GCs compared to placebo, although the number of ventilation-free days was still significantly increased [10]. It is noteworthy that this study even suggested an enhanced death rate due to neuromuscular weakness, possibly compensating for the beneficial effects on the lung. In contrast, a more recent meta-analysis provided good evidence that prolonged MP treatment rather accelerated the resolution of ALI and thereby significantly decreased mortality [9]. Along the same line, COVID-19 patients requiring mechanical ventilation were shown to benefit from Dex therapy [12]. Nevertheless, clinical evidence remains somewhat unconclusive [42].

The treatment of ALI patients at clinics is generally achieved by applying an initial dose of 2 mg/kg MP followed by tampering [9,10]. While doses ranging from 1 to 5 mg/kg of different GC derivatives have also been used in animal models [26,27], we administered 10 mg/kg BMZ or BNPs. The employed dose is thus somewhat higher than in humans and the potency of BMZ is greater than that of MP [43]. However, pharmacokinetics differs between mice and men, and therefore, drug doses cannot be compared one-to-one between species. Most importantly, however, the chosen therapeutic regimen has already been used in other mouse models of inflammatory diseases (experimental autoimmune encephalomyelitis, graft-versus-host disease), where an efficient improvement in clinical symptoms was achieved [35,36]. To allow for a direct comparison between these experimental approaches, we also employed the previously established dose in the ALI model at hand.

In this study, we reached out to compare the efficacy of GC therapy using BMZ in its traditional free form or encapsulated into IOH-NPs. GCs are lipophilic compounds that are distributed throughout the body via the circulation and can reach all cell types [44]. Hence, their activity is exerted not only in cells that are crucial for ALI pathology, but also other leukocyte subsets, as well as many cell types responsible for adverse effects. In contrast, GCs encapsulated into IOH-NPs are selectively engulfed only by a subset of cell types, while others respond to a far lesser extent. It is against this background that we speculated that targeted delivery of GCs using the new nanoformulation might improve ALI therapy. Our results now unveiled that BMZ and BNPs are similarly capable of mitigating hypothermia induced by sepsis, a major predisposing factor of ALI, and that they also both prevented airway inflammation and the systemic cytokine storm. Hence, the encapsulation of GCs into IOH-NPs did not compromise their therapeutic efficacy, albeit BNPs were unable to restore alveolar–capillary barrier integrity. It appears that endothelial as well as AT I cells in the lung are refractory to BNP action due to their inability to engulf them via micropinocytosis. In contrast, nanoformulated GCs have the capacity to target other pulmonary cell types with high efficacy, which explains why BNPs mitigate ALI to a similar extent as free BMZ. This notion is supported by the analysis of genetically engineered mice lacking the GR either in myeloid cells such as MΦ, MO and neutrophils, or selectively in AT II cells. Airway inflammation was refractory to BNP action in both mutant mouse strains, which suggests that GCs encapsulated in IOH-NPs achieve therapeutic efficacy at least partially by targeting the respective cell types.

An ideal therapeutic regimen should not only be efficient, but also devoid of adverse effects [45,46]. BNPs seem to partially fulfill this requirement. When we investigated the leukocyte composition in the blood, BMZ increased neutrophil numbers while reducing MO as well as T and B cells. In contrast, BNPs solely affected the frequencies of myeloid cells, while relative T and B cell numbers were unaltered. This finding indicates that antibody and T-cell effector functions might be retained after BNP therapy, which can be expected to facilitate the resolution of bacterial and viral infections, the major predisposing factors of ALI. The most devastating adverse effect of GCs in ALI patients, however, is myopathy [47]. It is believed that this disorder could explain why in some studies, the overall survival of ALI patients did not improve after GC therapy despite an increase in ventilation-free days [10]. Non-inflammatory myopathy involves catabolic and anti-anabolic mechanisms, including enhanced proteolysis, induction of myocyte apoptosis and inhibition of amino acid transport [48]. Our results show that GC delivery using IOH-NPs has the capacity to prevent the induction of muscle atrophy. Assuming that a similar effect can be observed in patients, this feature should improve the tolerability of GC therapy.

Our results uncover that the therapeutic efficacy of GCs is fully retained in a mouse model of ALI after their encapsulation into IOH-NPs accompanied by an increased cell-type-specificity, leading to reduced side-effects. This finding justifies further investigations into the suitability of this new therapeutic regimen in human patients.

## 4. Materials and Methods

### 4.1. Animal Experimentation

*Nr3c1^tm2Gsc^Lyz2^tm1(cre)lfo^* (GR^lysM^) on a BALB/c background, *Nr3c1^tm2GSc^Sftpc^tm1(cre/ERT2)Blh^* (GR^spc^) on a C57BL/6 background, the respective *Nr3c1^tm2Gsc^* (GR^flox^) littermate controls and C57BL/6 wildtype mice were bred at our animal facility at the University Medical Center Göttingen [49,50]. GR^flox^ mice contain two loxP sits flanking exon 3 of the GR (*Nr3c1*) gene in their genome. GR^lysM^ mice additionally express Cre under the control of the lysozyme M (*Lyz2*) promoter, whereas GR^spc^ mice additionally express an inducible Cre under the control of the surfactant protein C (*Sftpc*) promoter [21,51]. Gene recombination in GR^spc^ mice was induced with the help of tamoxifen (Sigma-Aldrich, Taufkirchen, Germany), which was dissolved in EtOH and sunflower oil at a 1:20 ratio and administered using an oral gavage at a concentration of 20 mg/mL in 150 µL thrice every other day.

Sepsis was induced by an administration of lipopolysaccharide (LPS), whereas it was combined with oleic acid (OA) to induce acute lung injury (ALI), as described [27,52]. Mice were injected with 10 mg/kg LPS (E.coli 055:B5) i.p., and if applicable, 2.6 mL/kg OA was administered i.v. 30 min later (both from Sigma-Aldrich). OA was prepared as a 4% solution in 0.1% BSA. Some mice received BMZ or BNPs 45 min after the initiation of the experiment, both at a dose of 10 mg of drug per kg body weight. During the last 3 h of every ALI experiment, the health status of the mice was continuously monitored to avoid excessive suffering. According to animal welfare regulations, all mice were sacrificed after a total of 15 h when full clinical symptoms had developed, but before exceeding the maximal tolerable level. The body weight loss was calculated relative to the average body weight during the two preceding days. Blood glucose concentration was measured with an Ascensia Blood Glucose Meter CONTOUR^®^ (Bayer, Leverkusen, Germany), and the body temperature was determined with a BIO-TK9882 thermometer in combination with a BIO-BRET-3 rectal probe (Bioseb, Vitrolles, France). Muscle atrophy was induced by treating C57BL/6 mice via daily i.p. injections of BMZ or BNPs for three consecutive days at a dose of 10 mg/kg body weight. The PBS injection served as the control. Mice were sacrificed one day after the last treatment, the skin was removed from the hind limbs to expose the muscle, the tibialis anterior and gastrocnemius were dissected, and the muscles weighed.

### 4.2. Flow Cytometric Analyses

The bronchoalveolar lavage fluid (BALF) was collected from the sacrificed mice via cannulation of the trachea and then flushing the lung thrice with 0.1% BSA. Cell counts were determined with the help of a Neubauer hemocytometer. Blood was obtained by heart puncture. Before flow cytometric analysis [53], the samples were treated with erythrocyte lysis buffer (20 mM Tris/HCl, 155 mM NH_4_Cl, pH = 7.2) followed by an FcR blockade using TruStain fcX (anti-mouse CD16/32; clone: 93). Cell suspensions were stained with monoclonal antibodies directly conjugated to different fluorochromes: anti-CD3ε (17A2), anti-B220 (RA3-6B2), anti-CD11b (M1/70), anti-F4/80 (BM8), and anti-Ly6G (1A8). A FACSCanto II flow cytometer (BD Biosciences, Heidelberg, Germany) was used for analysis in combination with FlowJo^®^ software (version 10.7.0; Treestar, Ashland, OR, USA). All antibodies were purchased from BioLegend (Uithoorn, The Netherlands). The gating strategy for the BALF analysis is depicted in Appendix A.

### 4.3. Histology

The lung was fixed in 4% Roti–Histofix (Carl Roth, Karlsruhe, Germany), followed by dehydration and embedding in paraffin. Subsequently, 2 µm thick sections were stained with hematoxylin and eosin (H&E) according to standard protocols. All images were acquired using a BX41 microscope (Olympus, Hamburg, Germany).

### 4.4. Quantitative RT-PCR Analysis

The lung was flushed with PBS/ 0.1% BSA before dissection in order to remove infiltrating leukocytes. Subsequently, RNA was isolated using the RNeasy^®^ Mini Kit (Qiagen, Hilden, Germany) and reverse-transcribed into cDNA with the iScript™ Kit (Bio-Rad, Munich, Germany), according to the manufacturers’ instructions. For the relative quantification of gene expression, an RT-qPCR reaction was performed employing the ABI 7500 Real Time PCR System (ThermoFisher, Darmstadt, Germany) and the Power SYBR^®^ Green Master Mix from the same company. Gene expression was determined with the ΔΔCt method using *Hprt* as the housekeeping gene. The sequences of the primers are as follows: *Hprt* (5′-GTC CTG TGG CCA TCT GCC TA-3′ and 5′-GGG ACG CAG CAA CTG ACA TT-3′), *Il1b* (5′-CTG ATC TGG GAT CCT CTC CA-3′ and 5′-AAG CAG CCC TTC ATC TTT TG-3′), *Il6* (5′-ATG TGC CTT CTT GGG ACT GA-3′ and 5′-CAG AAT TGC CAT TGC ACA AC-3′), *Tnf* (5′-ATG GCC TCC CTC TCA TCA GT-3′ and 5′-CTT GGT GGT TTG CTA CGA CG-3′), and *Ccl2* (5′-AGC ACC AGC CAA CTC TCA CT-3′ and 5′-CGT TAA CTG CAT CTG GCT GA-3′).

### 4.5. ELISA and Griess Test

The serum was prepared from the blood samples obtained via cardiac puncture. The levels of IL-1β, IL-6, TNFα and CCL2 were quantified via ELISA, using commercially available kits (BioLegend) according to the instructions of the manufacturer. The developing color was measured employing spectrophotometry at 450 and 570 nm using a Power Wave 340 microplate reader (BioTek Instruments, Wetzlar, Germany). NO_2_^−^ levels in the serum were evaluated using the Griess test and standards prepared from a NaNO_2_ stock solution. Of the diluted serum samples or standards, 50 μL was added to a 96-well plate and incubated with 50 μL 1% sulphanilamide in 2.5% phosphoric acid for 5 min at room temperature. Thereafter, 50 μL N-Naphtyl-ethylen-diamin-dihydrochlorid was added and again incubated for 5 min. Eventually, the absorption was measured at 540 nm on a microplate reader.

### 4.6. Alveolar–Capillary Barrier Permeability Test

C57BL/6 mice were i.v.-injected with Evans Blue (EB; Sigma) in PBS at a concentration of 25 mg/kg body weight 14 h after ALI induction [54]. One hour later, the mice were euthanized via an i.p. injection of Ketamin and Xylariem. As soon as the reflexes of the mice were no longer observed, the thoracic cavity was opened and an injection needle inserted into the right ventricle. The left atrium was punctured and 20 mL of ice-cold PBS/ 5 mM EDTA was flushed through the cardiovascular system. The lung was dissected and weighed, and 4 mL formamid per g of lung tissue was added. After incubation at 60 °C overnight, the samples were centrifuged at 5.000× *g* for 30 min at room temperature and the supernatant was collected. The absorbance was measured at 620 and 740 nm using a microplate reader and the concentration of EB in the lung was calculated using a standard curve and the following formula: E_620_ = E_620_ − (1.426 × E_740_ + 0.03).

### 4.7. Nanoparticle Synthesis and Characterization

IOH-NPs with the chemical composition [ZrO]^2+^[(BMP)_0_._9_(FMN)_0_._1_]^2–^ (designated BNPs) were prepared as previously described [33], and redispersed at a concentration of 2.8 mg/mL in H_2_O. The synthesis and characterization of BNPs are summarized in Appendix A.

### 4.8. Statistical Analysis

All data were analyzed via an unpaired one-way ANOVA, followed by the Newman–Keuls multiple-comparison test. Analyses were performed using GraphPad Prism^®^ software, version 9 (San Diego, CA, USA). Data are depicted as scatter dots plots, dot–box plots or box plots, with the mean ± SEM indicated in each graph. Levels of significance: n.s.: *p* > 0.05; *: *p* < 0.05; **: *p* < 0.01; ***: *p* < 0.001.

## Figures and Tables

**Figure 1 ijms-24-16843-f001:**
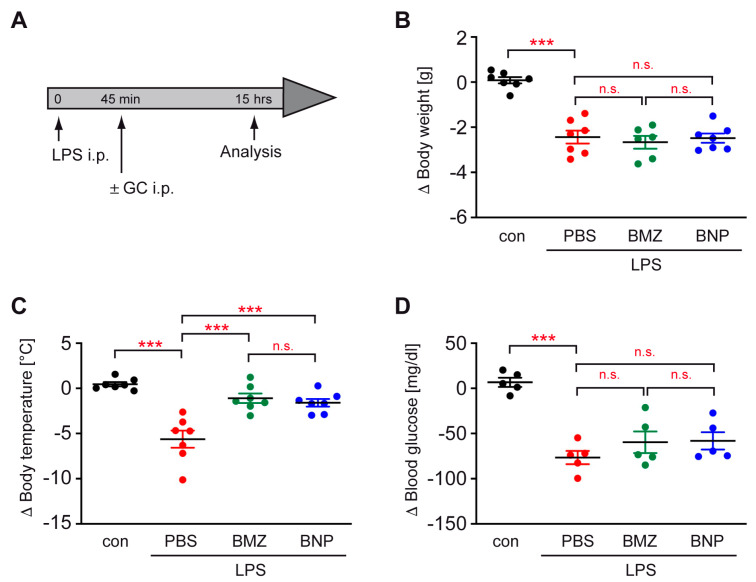
Efficacy of GC therapy in a mouse model of endotoxin-induced sepsis. (**A**) C57BL/6 mice were injected with LPS, and after 45 min, they received 10 mg/kg BMZ or BNPs containing the same amount of drug. PBS served as the control. After a total of 15 h, the mice were sacrificed and analyzed. (**B**) The absolute change in body weight (Δ) is depicted as the mean ± SEM (N = 7). (**C**) The body temperature was determined using a rectal probe and its change is depicted as the mean ± SEM (N = 7). (**D**) The change in the blood glucose level is depicted as the mean ± SEM (N = 5). Statistical analysis was performed via a one-way ANOVA, followed by the Newman–Keuls multiple-comparison test. Levels of significance: n.s.: *p* > 0.05; ***: *p* < 0.001.

**Figure 2 ijms-24-16843-f002:**
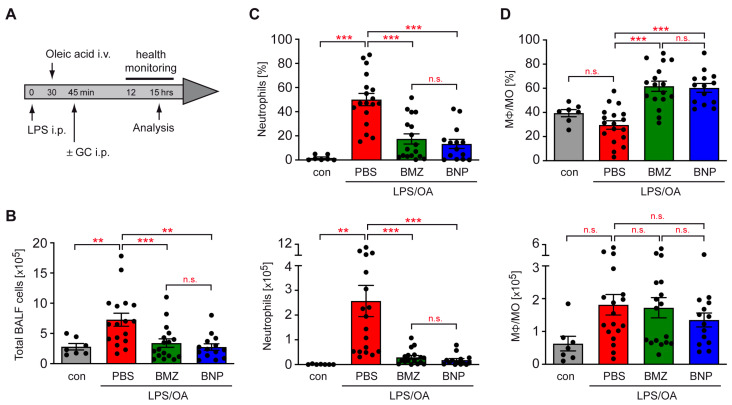
Repression of airway inflammation by GCs in a mouse model of ALI. (**A**) C57BL/6 mice were injected with LPS, and 30 min later, they were treated with OA. After another 15 min, the mice received 10 mg/kg BMZ or BNPs containing the same amount of drug. PBS served as the control. The mice were sacrificed after a total of 15 h and the lung was flushed with PBS. (**B**) Absolute numbers of BALF cells in a volume of 1 mL are depicted as the mean ± SEM (N = 7–17). (**C**,**D**) BALF cells were stained with monoclonal antibodies against CD11b, Ly6G and F4/80 to distinguish neutrophils from MΦ/MO. Analysis was achieved via flow cytometry. Percentages and absolute cell numbers in 1 mL are depicted as the mean ± SEM (N = 7–18). Statistical analysis was performed using a one-way ANOVA, followed by the Newman–Keuls multiple-comparison test. Levels of significance: n.s.: *p* > 0.05; **: *p* < 0.01; ***: *p* < 0.001.

**Figure 3 ijms-24-16843-f003:**
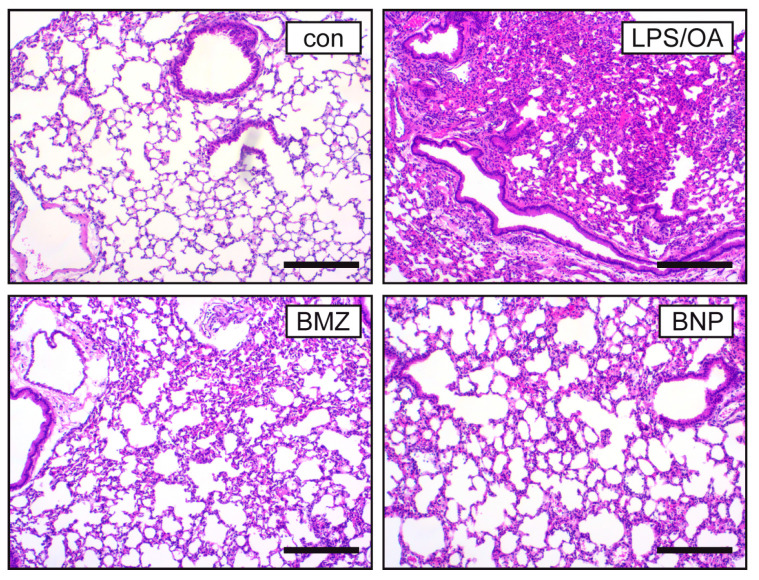
Histological analysis of the lung in a mouse model of ALI. C57BL/6 mice were injected with LPS and OA followed by treatment with BMZ or BNPs according to the experimental setup outlined in Figure 2A. Mice receiving only PBS served as the control (con). The lung was fixed in PFA followed by H&E staining of pulmonary structures and leukocyte infiltration. One representative section out of 4–7 mice in total is depicted for each condition. The size bar corresponds to 500 μm.

**Figure 4 ijms-24-16843-f004:**
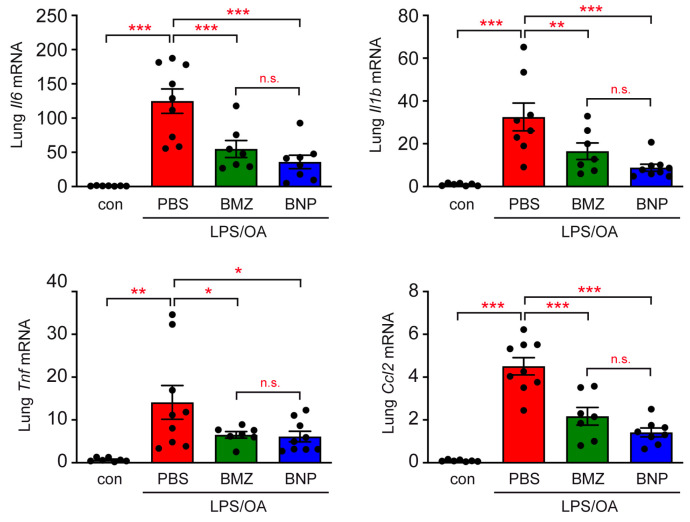
Gene expression analysis of the lung in a mouse model of ALI. Airway inflammation was induced in C57BL/6 mice with the help of LPS and OA, followed by treatment with BMZ or BNPs according to the experimental setup outlined in Figure 2A. Mice receiving only PBS served as the control. After 15 h, total RNA was isolated from the lung and reverse-transcribed into cDNA. RT-qPCR analysis was used to determine the relative gene expression of *Il6*, *Il1b*, *Tnf*, and *Ccl2* via normalization to the housekeeping gene *Hprt*. Gene expression in the control mice was arbitrarily set to 1 and is depicted as the mean ± SEM (N = 7–9). Statistical analysis was performed using a one-way ANOVA, followed by the Newman–Keuls multiple-comparison test. Levels of significance: n.s.: *p* > 0.05; *: *p* < 0.05; **: *p* < 0.01; ***: *p* < 0.001.

**Figure 5 ijms-24-16843-f005:**
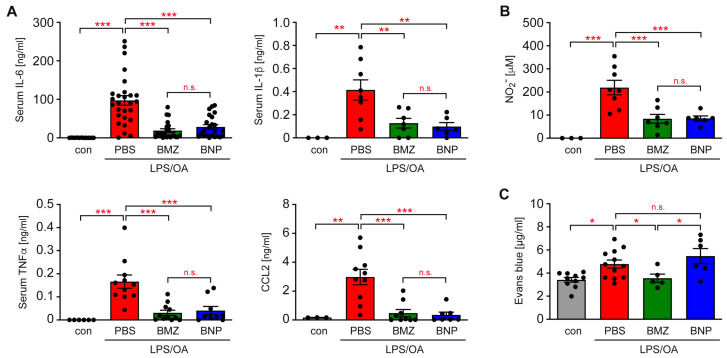
Analysis of inflammatory mediators in the serum and the permeability of the alveolar–capillary barrier in a mouse model of ALI. (**A**) C57BL/6 mice were injected with LPS and OA, followed by treatment with BMZ or BNPs according to the experimental setup outlined in Figure 2A. Mice receiving only PBS served as the control. Blood was collected by heart puncture and cytokines were determined in the serum via ELISA. Concentrations of IL-6, IL-1β, TNFα and CCL2 are depicted as the mean ± SEM (N = 6–27). (**B**) The concentration of NO_2_^−^ in the serum was analyzed using the Griess reagent, followed by spectrophotometric analysis. Absolute concentrations were determined with the help of a standard curve and are depicted as the mean ± SEM (N = 6–9). (**C**) Alveolar–capillary barrier permeability was measured via an i.v. injection of Evans Blue (EB) dye and its subsequent quantification in lung tissue using a spectrophotometric test. The concentration of EB in the elute from the lung is depicted as the mean ± SEM (N = 5–10). The higher the concentration of EB, the leakier the alveolar–capillary barrier is. Statistical analysis in all panels was performed via a one-way ANOVA, followed by the Newman–Keuls multiple-comparison test. Levels of significance: n.s.: *p* > 0.05; *: *p* < 0.05; **: *p* < 0.01; ***: *p* < 0.001.

**Figure 6 ijms-24-16843-f006:**
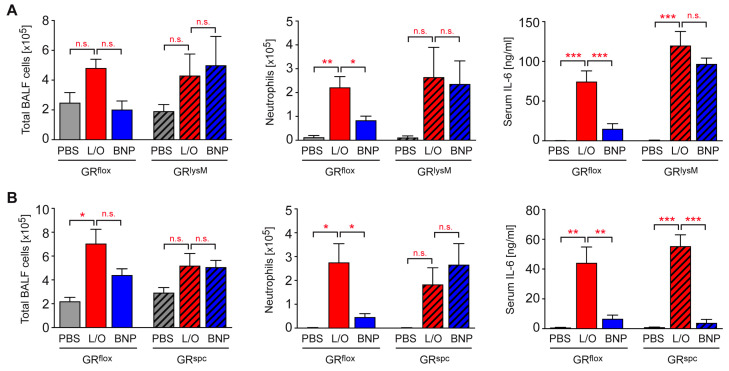
Repression of airway inflammation by GCs in mice with cell-type-specific disruptions of the GR. (**A**) ALI was induced in GR^flox^ control mice as well as GR^lysM^ mice lacking the GR in myeloid cells by combined treatment with LPS and OA. Some mice additionally received BNPs, and mice injected only with PBS served as the control. (**B**) ALI was induced in GR^flox^ control mice as well as GR^spc^ mice lacking the GR in AT II cells. Some mice additionally received BNPs, and mice injected only with PBS served as the control. Total cell numbers as well as the numbers of neutrophils in a volume of 1 mL of BALF were enumerated using a Neubauer hemocytometer and flow cytometric analysis. The concentration of IL-6 in the serum was determined by ELISA. All values are depicted as the mean ± SEM (N = 3–8 for GR^lysM^ mice, and N = 4–11 for GR^spc^ mice). Statistical analysis in all panels was performed with a one-way ANOVA followed by the Newman-Keuls multiple comparison test. Levels of significance: n.s.: *p* > 0.05; *: *p* < 0.05; **: *p* < 0.01; ***: *p* < 0.001.

**Figure 7 ijms-24-16843-f007:**
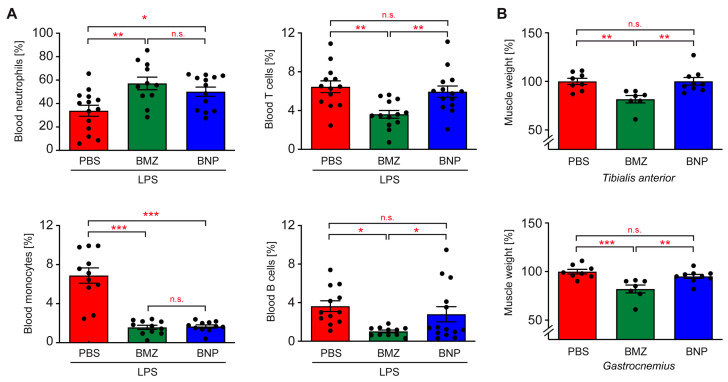
Evaluation of adverse GC effects in mice. (**A**) C57BL/6 mice were injected with LPS, and 45 min later, they were treated with 10 mg/kg BMZ or BNPs containing the same amount of drug. Mice receiving PBS served as the control. After 15 h, the mice were sacrificed and the blood samples were stained with monoclonal antibodies, followed by flow cytometric analysis (T cells: CD3^+^, B cells: B220^+^, neutrophils: CD11b^+^Ly6G^+^, monocytes: CD11b^+^Ly6G^−^F4/80^+^). Data are depicted as the mean ± SEM (N = 10–14). (**B**) C57BL/6 mice were injected daily with 10 mg/kg BMZ or BNPs containing the same amount of drug for three consecutive days. PBS served as the control. On the fourth day, the mice were sacrificed and the tibialis anterior and gastrocnemius muscles dissected. The mean weight of each muscle in the control mice receiving PBS was set to 100% and the muscle weights in the treated mice were calculated accordingly. Data are depicted as the mean ± SEM (N = 8/7/9). Statistical analysis was performed via a one-way ANOVA, followed by the Newman–Keuls multiple-comparison test. Levels of significance: n.s.: *p* > 0.05; *: *p* < 0.05; **: *p* < 0.01; ***: *p* < 0.001.

## Data Availability

Data and materials are available from the corresponding author upon reasonable request.

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
