# Peer review of "Glucocorticoid Nanoparticles Show Full Therapeutic Efficacy in a Mouse Model of Acute Lung Injury and Concomitantly Reduce Adverse Effects"

_ijms, 2023, doi:10.3390/ijms242316843_

Round 1

Reviewer 1 Report

Comments and Suggestions for Authors

Albers et al developed BNP for the ALI therapy and compared BNP to free betamethasone (BMZ) in their ability to suppress clinical symptoms and molecular hallmarks of ALI. Their findings is interesting. However, the followed two  issues should be address.

1.The characterization of BNP is missing in the manuscript.

2. How many BNP reached to the lung after i.p. should be described in the results.

Reviewer 2 Report

Comments and Suggestions for Authors

In the manuscript entitled “Glucocorticoid nanoparticles show full therapeutic efficacy in a mouse model of acute lung injury and concomitantly reduce adverse effects”, Albers and colleagues are examining the efficacy and tolerability of encapsulated betamethasone (BNP), compared to the free form of betamethasone (BMZ) in a mouse models of acute lung injury. Overall their findings support improved therapeutic potential for BNP over BMZ, with both forms showing similar reductions in inflammation, while BNP improved B and T cell numbers and reduced muscle wasting within 15 hours of ALI induction when compared to BMZ. While these results are promising, there are some concerns with the data presented which are outlined below. 

Major:

The concentration used of BMZ/BNP was 10 mg/kg, this seems quite high. A brief search of the literature reveals other similar ALI studies using lower concentrations even with less potent forms of glucocorticoids. Betamethasone is one of the most potent glucocorticoids, not to mention the BNP is thought to be a more targeted, cell specific approach which would indicate even higher amounts are getting into the cells of interest than the free form. I understand this provides the benefit of reducing off-target effects, but request that the authors provide rationale for using this dose for their studies and mention to the readers how this compares with current pharmacological approaches currently in use.

Comments regarding duration of assessment would also be beneficial as 15 hours is not very long to determine true efficacy on treatment outcomes. 

Some sort of quantification would be helpful for the histology data, if possible. In the absence of quantification please provide 2-3 slide images from different animals from each group to support that these are in fact representative of the groups. Please also include the scale bar within each image.

Please state why BMZ was not utilized as a comparison group for the cell-specific GR knockout studies and, if appropriate, whether you think it would yield similar findings.

Please provide more information and rationale for the examining GC activity AT II cells. There is no mention in the discussion for any of the data in Figure 6.

Minor:

Citations needed for lines 60-61

Some of the text in the results section would be more appropriate in the discussion, such as for lines 197-199.

Please provide clarification/greater detail for which cell surface markers you are using to identify and sort amongst neutrophils, macrophages and monocytes.

Comments on the Quality of English Language

Minor grammar and spelling corrections needed, otherwise reads well. 

Round 2

Reviewer 1 Report

Comments and Suggestions for Authors

accept

Reviewer 2 Report

Comments and Suggestions for Authors

I appreciate the authors addressing all concerns from the initial review with appropriate responses and edits.